# Development of an Artificial Intelligence-Based Breast Cancer Detection Model by Combining Mammograms and Medical Health Records

**DOI:** 10.3390/diagnostics13030346

**Published:** 2023-01-17

**Authors:** Nguyen Thi Hoang Trang, Khuong Quynh Long, Pham Le An, Tran Ngoc Dang

**Affiliations:** 1Department of Biomedical Engineering, College of Engineering, National Cheng Kung University, Tainan 701, Taiwan; 2Center for Population Health Science and Data Science, Ha Noi University of Public Health, Ha Noi 100000, Vietnam; 3Family Medicine Training Center, University of Medicine and Pharmacy at Ho Chi Minh City, Ho Chi Minh City 700000, Vietnam; 4Grant and Innovation Center, University of Medicine and Pharmacy at Ho Chi Minh City, Ho Chi Minh City 700000, Vietnam; 5Department of Environmental Health, Faculty of Public Health, University of Medicine and Pharmacy at Ho Chi Minh City, Ho Chi Minh City 700000, Vietnam

**Keywords:** breast cancer, Xception, Resnet-v2, Resnet50, VGG16, CNN, k-nearest neighbor, support vector machine, random forest, artificial neural network, gradient boosting machine

## Abstract

Background: Artificial intelligence (AI)-based computational models that analyze breast cancer have been developed for decades. The present study was implemented to investigate the accuracy and efficiency of combined mammography images and clinical records for breast cancer detection using machine learning and deep learning classifiers. Methods: This study was verified using 731 images from 357 women who underwent at least one mammogram and had clinical records for at least six months before mammography. The model was trained on mammograms and clinical variables to discriminate benign and malignant lesions. Multiple pre-trained deep CNN models to detect cancer in mammograms, including X-ception, VGG16, ResNet-v2, ResNet50, and CNN3 were employed. Machine learning models were constructed using k-nearest neighbor (KNN), support vector machine (SVM), random forest (RF), Artificial Neural Network (ANN), and gradient boosting machine (GBM) in the clinical dataset. Results: The detection performance obtained an accuracy of 84.5% with a specificity of 78.1% at a sensitivity of 89.7% and an AUC of 0.88. When trained on mammography image data alone, the result achieved a slightly lower score than the combined model (accuracy, 72.5% vs. 84.5%, respectively). Conclusions: A breast cancer-detection model combining machine learning and deep learning models was performed in this study with a satisfactory result, and this model has potential clinical applications.

## 1. Introduction

Breast cancer is one of the high-prevalence cancer types among women, accounting for 12.5% of annual new cancer cases worldwide. The International Agency for Research on Cancer (IARC) estimated that there were about 2.26 million new cases of breast cancer and approximately 685,000 deaths worldwide in 2021. One of the main issues with this disease is late detection, which is an important factor in reducing survival rates. It is estimated that the average 5-year survival rate for women with localized cancer is 97.5%, whereas the 5-year survival rate for breast cancer that has spread to a distant part of the body is 29% [1]. To reduce the mortality rate, early detection with adjunct methods in clinical assessment has received attention in recent years. Mammography examination is the main imaging method of breast cancer screening in asymptomatic patients and has been shown to be effective in reducing mortality rates by 30–70% [2]. In clinical diagnosis, mammograms are read and classified by a radiologist. The findings are reported according to the Breast Imaging Reporting and Data System (BI-RADS) score [3]. A finding of an abnormal area on a mammography image will require more tests, such as special mammogram views or ultrasonography. A further test with biopsy is considered if these findings are suspicious for cancer. Analyzing these images, however, is difficult because of various lesion types and the difference between lesions and dense breast tissue. In addition, density tissue can cover malignancy lesions, decreasing the mammogram’s sensitivity [4]. As a beneficial and necessary computerized image technique for breast cancer detection, computer-aided diagnosis (CAD) can improve the detection of breast cancer and provide a second opinion to support radiologists in detecting lesions and making diagnostic decisions [5,6]. Furthermore, it can estimate the likelihood that a lesion is benign or malignant. The CAD system is based on several processes: preprocessing, image augmentation, feature extraction, feature selection, and model classification. 

### 1.1. Related Work

Deep learning (DL) approaches have been widely applied to analyze medical imaging as a reliable technique to learn features from original images automatically [7]. Several studies have performed DL algorithms for mammography image classification using publicly available datasets. Dhabyani et al. implemented training on the BUSI dataset to evaluate and compare the performance of different DL models such as Alexnet, Resnet, VGG16, Inception, and NASNet to classify breast tumors. The best result was obtained from an Inception network with 94% accuracy [8]. The training of DL algorithms usually requires a large amount of data, which can be a limitation in breast cancer studies due to the lack of diverse datasets or the small number of images in the dataset [9,10]. Transfer learning (TL) is a powerful technique for training a small dataset without overfitting. TL is employed by pre-training a deep neural network on a large dataset, and then fine-tuning to make them more relevant for a specific task. A deep pre-trained neural network such as the Resnet50, VGG16, VGG19, and Interception-v2 for classifying breast tumors in mammograms was proposed by Saber et al. [10]. They applied deep learning approaches and transfer learning techniques on a small dataset to achieve the best result with 96% accuracy. All studies above suggested that the deep neural network model for transfer learning obtained high detection capabilities. Therefore, using these models can be an efficient tool when the target dataset is substantially small.

With the development of automated breast cancer detection systems, many recent studies have been deployed using enhanced deep neural network models [11,12,13,14,15]. Chakravarthy et al. [12] proposed an optimization technique for breast cancer detection. In this study, a customized method of integrating a ResNet18 model with the extreme learning machine (ELM), optimized using a Crow-Search (ICS) algorithm, obtained a significant improvement in accuracy with 97.2% for DDSM, 98.1% for MIAS, and 98.3% for INbreast datasets. An ensemble technique for classifying breast cancer from mammography images was proposed by Altameem et al. [13]. The authors employed a fuzzy rank-based Gompertz function to incorporate the best features of different deep CNNs and create final predictions. This algorithm outperforms each CNN model with 99.3% accuracy. Muduli et al. [14] developed a detection method using a CNN model to learn discriminant features automatically and classify breast cancer based on mammogram and ultrasound images. The performance achieved an accuracy of 96.5% on MIAS and 100% on BUS-1 datasets. In addition, many deep learning algorithms on breast cancer were developed using histological images [16,17]. Wakili et al. [17] proposed a novel neural network model namely DenTnet which took the benefits of both DenseNet and transfer learning. This model demonstrated better accuracy over transfer learning methods on the same datasets.

Previous studies showed that feature extraction is one of the essential steps in building efficient machine-learning (ML) models to identify benign and malignant tumors [18]. They extracted a subset of features from the lesion region on mammograms, such as breast density, the contour of a mass, distorted structures, calcifications, or tumor shape. Daniel et al. presented image descriptors including intensity, texture, multi-scale texture, and spatial distribution of the gradient for breast cancer diagnosis [19]. These features are computed from a lesion region on mammograms and trained with ML classifiers. Maha et al. proved that various features could affect the performance results [18]. In this study, the mammogram features were extracted into three groups: intensity-based, shape-based, and texture-based. They trained an optimized support vector machine (SVM) and Naïve Bayes after employing a feature selection and hyperparameter optimization schemes to achieve an accurate model. Delen et al. [20] proposed binary classifiers (SVM, Random Forest, and Logistic Regression) based on clinical characteristics and gene expression for breast cancer prognosis. They used an available dataset including 200,000 samples for evaluating the model. Three models achieved 93.6%, 91.2%, and 89.2% accuracy, respectively. In a Burke et al. study [21], different models including PCA, Decision Tree (DT), and ANN were trained on 8271 samples. The result obtained AUC in a range from 0.71 to 0.78, and the best-reported model is an ANN. Previous studies showed that the most extracted features are handcrafted features, and the number of these features can reach hundreds of thousands. However, it may not fully represent the specific lesion in the tumor. Moreover, extracting the breast image descriptors requires a good understanding of the tumor from radiologists [22]. Therefore, there can be a significant impact on classification performance.

On the other hand, we investigated that breast cancer prediction models based on clinical data can support physicians in evaluating the capability of a woman to develop breast cancer. In Catherine et al.’s research, a prediction model for breast cancer in the general women’s population by using risk factors and clinical assessments was reported [23]. This model showed a 0.76 AUC value and 95% confidence interval (95% CI: 0.70, 0.82).

DL approaches have exhibited the potential to deal with a small dataset. However, these algorithms usually lack interpretability, hence combining DL with clinical variables can clarify the result obtained. Therefore, a combined model using mammograms and clinical variables based on DL and ML approaches was taken as the main research aim of this study. In this work, we investigate whether adding more clinical information to the mammogram-based model can improve the performance, compared to either the image model or the clinical model only, in estimating the probability of breast cancer. Additionally, it is necessary to consider whether the same algorithm is applied to various data types or whether different algorithms yield better results for different data types.

### 1.2. Novelty and Contribution

According to a survey of the relevant literature, researchers only worked on a single DL or ML model for classifying breast images and few studies applied risk factors and clinical assessments into the detection model. Nevertheless, merging breast images with clinical features may boost the performance and robustness of a detection system. In addition, to the best of our knowledge, the creation of an ML-DL model training on a dataset of linked mammograms and health records was not reported in previous studies. Therefore, we developed a combined model to investigate this problem. Our findings might be useful for improving the accuracy of cancer detection and we believe that the use of this model as a second reader could be beneficial.

The main contributions of this paper are summarized as follows:We have proposed an AI framework based on ML-DL approaches which include various algorithms for each data type. Moreover, this study attempts to use mammography images combined with clinical variables as input for breast cancer detection;Multiple deep learning models to detect breast cancer in mammograms, including X-ception, VGG16, ResNet-v2, ResNet50, and CNN3 were employed. An augmentation technique was utilized for creating more training samples to avoid overfitting;To determine the most common clinical features related to cancer capability and select an appropriate ML model based on levels of model complexity to achieve high accuracy and expedite the learning process;We have developed an effective model combination for breast cancer detection based on the mammogram and clinical features to comprehensively assess at the individual patient level.

## 2. Materials and Methods

### 2.1. Study Design and Data Preparation

This cross-sectional study comprised 357 women (136 malignant, 221 benign) collected from the Oncology Hospital Ho Chi Minh City between July 2017 and September 2017. The dataset contained information obtained from patients who underwent at least one mammogram examination, all clinical variables were recorded by physicians, and all mammography findings were made by radiologists. We excluded the patients with the following criteria: a history of breast cancer, previous breast cancer treatment (operation, chemotherapy, or radiation), and mammograms that were BI-RADS category 0–1. From 357 patients obtained, 731 mammography images were labeled as a binary class based on the biopsy result (benign or malignant). These mammography images were in JPEG format with three channels (red, green, and blue). The original size of the images was 3328 × 2560 pixels and the majority of mammogram pixels were background pixels which did not add any contribution to breast cancer detection. Hence, background removal was completed. All patients were in the age range 48 ± 11.4 years old, with a minimum of 19 and a maximum of 90.

There were 24 variables in the clinical dataset extracted from each patient (Appendix A Table A1). These features were associated with information about the patient, risk factors, symptoms, ultrasound which describes the feature of the lesion, and mammogram findings. There were some missing values in the dataset, which is indicated by the text *NaN*. All patients were split into a training set (80% of the original dataset, 286 subjects) for constructing the model and a testing set (a remaining subject which is 20% of the dataset) for evaluating the model’s performance.

### 2.2. Breast Cancer Detection Algorithm

This study proposes a breast cancer detection algorithm using machine learning (ML) and deep learning (DL) frameworks. Figure 1 shows the proposed algorithm flowchart. For each woman, the input data were the detailed clinical variables and mammography images in two views: craniocaudal (CC) and mediolateral oblique (MLO). The deep-learning framework consists of four transfer learning models: X-ception, VGG16, Resnet-v2, and Resnet50; a three layer CNN (CNN3) model is trained on mammography images to evaluate and compare the performance in detecting breast cancer. We selected the most important clinical features having the highest contribution to discriminating between benign and malignant. Five general machine learning classifiers, including k-nearest neighbor (KNN), support vector machine (SVM), artificial neural network (ANN), random forest (RF), and gradient boosting machine (GBM) were implemented to obtain the classification results. Finally, when analyzed for each woman, a combination model was evaluated by integrating clinical features with mammograms. The final probability for benign or malignant discrimination was estimated from the average probability of the ML-DL model.

### 2.3. Deep-Learning Classifiers

#### 2.3.1. Data Augmentation

The original mammography images have a large size that may affect classification performance and are also time-consuming. Thus, the converted images were resized to match the specification input of particular models. For each Xception, VGG16, Resnet-v2, Resnet50, and CNN model, the input is a 3D RGB (three-dimensional red, green, and blue) image with a size of 299 × 299 × 3.

Due to the small number of mammogram images, data augmentation techniques have been applied to increase the original dataset size and improve the performance of the model. This study employed geometric transformations such as rotation, horizontal flipping, zooming, and scaling. The transformed images were reflected and rotated by different degrees to recognize the key detection properties of breast cancer. Some previous studies [24,25,26] proposed the impact of data augmentation techniques that aimed to improve performance by expanding their training set.

#### 2.3.2. Deep Neural Networks

The model used for mammography images was built using deep neural network algorithms. Due to the dataset being comparably small, transfer learning was adopted as an efficient tool to enhance performance and use less computational time [27]. In this study, four different models were proposed and compared for the identification of benign and malignant, including three transfer learning models: Xception, VGG16, Resnet-v2, Resnet50, and a CNN3 model.

Xception is a convolutional neural network architecture that assumes that the entry channels of cross-correlation with spatial correlation in the feature maps are completely separate [28]. Xception is extended from the Interception architecture with 36 convolutional layers replacing the traditional convolution layer with the depthwise convolution layer with residual connections. This network performs better than the InterceptionV3 on the ImageNet dataset for classification tasks at the same number of parameters [29].

One of the most common deep neural networks is VGG16 which has simple layers. The well-known architecture for VGG16 has included 41 layers with a small 3 × 3 kernel on all convolutional layers. In previous studies, VGG16 also utilized the transfer learning technique to extract high-level features from the original image and perform classification tasks. It showed a reduced error rate as well as training time for the classification of breast cancer [30].

Resnet-v2 is an Interception network based on the computational cost of the Interception-v4. This model is trained on the ImageNet dataset to classify 1000 classes with a top one error rate of 3.5% [31]. This architecture’s main benefits are reducing dimensions while keeping a lot of information about the relative feature input without the computational complexity of similar networks [32]. 

Resnet50 is a convolutional neural network that includes 50 deep layers (48 Convolution layers along with 1 Max pooling and 1 Average Pool layer) with the residual block. This model has over 23 million trainable parameters indicating a deep architecture that makes it better for image classification. The strength of this concept is to skip connections and pass the residual to the next layer so that the model can continue to train, which relies on the core of the residual blocks [33]. Thus, ResNet improves the efficiency of deep neural networks while minimizing errors.

A traditional deep network algorithm of three convolutional layers and two fully connected (FC) layers, known as CNN3, was constructed. The input images were resized into 224 × 224. All convolutional layers used a 3 × 3 kernel followed by the max-pooling layers to generate the feature maps. Finally, a global average pooling was applied in the first FC to reduce the number of parameters, and the output layer indicated probabilities of benign and malignant. These layers have been demonstrated experimentally to build a sufficient CNN model, which was also executed in previous studies [34,35].

In the training of deep learning models, the original model’s architecture has been preserved excluding the layers that come after the convolutional base. The weights of the convolutional layers were frozen, and three fully connected layers were added according to each deep CNN model. In this study, we used the same set of parameters to train all deep CNN models on the mammogram dataset. The parameter values such as the learning rate, number of epochs, and batch size were set to *0.00001, 30, and 32*, respectively. Moreover, we utilized a RMSProp (root mean square propagation) optimizer to minimize the loss function with *decay = 0.9* and *epsilon = 1e*−*8*. Finally, the last layer used a softmax activation function to classify the image into two classes (benign and malignant). The output of this layer estimated a probability distribution of the predicted classes.

### 2.4. Machine-Learning Classifiers

#### 2.4.1. Data Preprocessing

In this study, data preprocessing consisted of three steps. The first step in this stage is the treatment of missing data. Multivariate Imputation by Chained Equations (MICE) was implemented to deal with this problem. In this study, MICE assumes that the missing data are Missing At Random (MAR), and then these were estimated with the value calculated using the weighted average of the non-missing values, where the weight is approximate with the values of the nearest matrix from the known data. 

Second, we removed near-zero variance features because they are almost constant and have less predictive power. We also excluded variables that were highly correlated with each other since these variables would be invalid in the results of ML models. Third, the normalization step was applied to rescale the data by using Z-Norm since most of the models were impacted by different scaling of the variables.

Finally, we defined a subset of variable importance from the initial 24 variables which were chosen by a feature selection method. This procedure was conducted to find the highest-ranking set of features and simplify the model complexity. In this study, the optimal set of features was selected by Recursive Feature Elimination (RFE). This method is implemented by eliminating recursive features and building a model on the remaining features to calculate the accuracy. RFE also computes an importance score for each feature as a contribution to the model.

#### 2.4.2. Model Description

In model comparison, we investigated five different machine learning algorithms to detect breast cancer. Our selection models were based on diversifying the choice of methodologies and the level of model complexity (decision tree, kernel method, and neural networks). A basic classification technique such as K-NN was selected to verify the complexity of the problem. We chose SVM with a Radial Basic Function (RBF) Gaussian kernel from the kernel approaches because it can handle data noise and nonlinearity. Furthermore, we considered a Neural Network model representing a significant class of non-linear predictive models. We considered RF and GBM algorithms from the decision tree methods since they are famous ensemble-based decision tree techniques.

The *k*-nearest neighbor (KNN) is a lazy learning method since learning only occurs when testing data that need to be classified. It computes the similarity or nearest distance between testing data with every data in the training set to decide the class of the new data. In the training phase, the *k*-closest data (*k*-nearest neighbors) are then chosen based on the minimum distance with unlabeled testing data and assigned to the class that was the most frequent class among the k-nearest neighbors. The important component of the KNN model is the distance function which can be determined by Euclidean, Minkowski, and cosine-distance metrics [36,37].

A support vector machine (SVM) is a supervised learning model used for classification and regression tasks. SVM depends on finding the best hyperplane to separate the features into different domains. In binary classification, it is assumed that p-dimensional space can be divided by (p-1) dimensional hyperplanes, and those hyperplanes separate the data point into their potential classes. The best hyperplane is the largest margin between two classes, and the closest data points on the hyperplane boundary are called support vectors [38].

An artificial neural network (ANN) is based on the structure of a simple multilayer perceptron model consisting of interconnected nodes. The input data are converted to output in each node and fed as input to the next layer. The ANN model was built as a three-layer feed-forward to simplify computationally. The layers include an input layer, a hidden layer, and an output layer with a single node. During the training phase, the weight of the model (*decay* hyperparameter) was tuned by increasing or decreasing the value of this weight. The *sigmoidal* function was used to connect hidden units. The output node generated values of 0 and 1 that showed the capability of malignancy [39]. 

Random forest (RF) is a popular machine-learning algorithm for classification tasks. It uses random subsamples of the training set to generate a large number of decision trees, each consisting of randomly varying features. RF presents an advantage on decision trees by using an ensemble technique to handle the sensitivity of decision trees. Finally, the result is calculated by averaging results from every tree in the forest [40].

A gradient boosting machine (GBM) is an ensemble forward learning technique used for regression and classification problems. It uses a decision tree as the base classifier to train the input data. This algorithm combines all weaker base classifiers and generates a strong predictive model. Afterward, a loss function is computed based on the difference between the actual and predicted value. Hyperparameters for each base classifier are adjusted by increasing or decreasing depending on the error value. Eventually, this process determines the best model with minimum training loss [41].

#### 2.4.3. Model Parameter Tuning

Data resampling techniques such as k-fold cross-validation, leave-one-out, or bootstrapping are used for model evaluation as well as tuning parameters. The parameters are fine-tuned by grid search, ensuring the model performance is more realistic [42]. In this approach, a set of possible tuning parameter values is defined. Then the dataset is split into the training set and testing set. An additional validation set is generated from the training set to determine the hyperparameter’s values that assess a model fit on the training dataset. The previous step was repeated for multiple iterations and calculated the average performance over all iterations. Finally, the optimal parameter is selected corresponding to the best model performance. 

In our study, we used the *k*-fold cross-validation approach to account for both parameter tuning and training phase validation. As shown in Figure 2, we divided the dataset into training and testing sets following a ratio of 8:2. This study used *k* = 5, then the training set was split consecutively into 5 sub-folds to ensure that there were the same data splits and repetition in model comparison. Subsequently, each sub-fold can be used as the validation set to calculate the average performance and determine the optimal parameter set with the remaining (*k*-1) sub-folds as the training set. Within each repetition, each hyperparameter was tuned in the training set through nested 5-fold cross-validation. The test set was used to evaluate the performance of the model based on selected hyperparameters. This procedure was repeated 10 times to yield a better estimate of the test set performance.

Table 1 particularizes these classifiers with model hyperparameters being fine-tuned by a grid search with a 5-fold cross-validation method to select the set of hyperparameters that achieve the best performance and avoid overfitting. In this study, a number of *k* as 10 was the optimal for the KNN model. In the SVM model with two parameters, the cost choice *C* and a kernel smoothing parameter *σ* of 1 were selected. The tuning parameters of the ANN model included the number of hidden layers optimized from 5 to 10, and weight decay was set to 0.1 and 0.5 for the training process to avoid overfitting. The *mtry* parameter was used as a tuning parameter for the RF model. For the GBM, an interaction depth of 1 and a number of trees of 50 were the best parameters; additionally, *shrinkage*, known as the learning rate, was considered to improve performance significantly.

### 2.5. Performance Evaluation Metrics

Evaluating a model is an important step in developing an effective classification model. There are several evaluation metrics, such as cross-validation, confusion matrix, the receiver operating characteristic (ROC) curve, and the area under ROC curve (AUC). The confusion matrix represents the instance used to calculate evaluation metrics: true positive (TP) is the number of correctly predicted malignant, while false positive (FP) is the number of incorrectly predicted malignant. Similarly, true negative (TN) reflects the number of exactly predicted benign, and false negative (FN) indicates the number of incorrectly predicted ones. The accuracy, sensitivity, and specificity are frequently used to evaluate the performance model. 

As performance metrics, we computed in terms of accuracy, sensitivity, and specificity for the test set. The ROC curve is a plot of FP against TP and estimates the area under the ROC curve (AUC). The formula of each metric can be calculated as:(1)$$\mathrm{Accuracy}=\frac{\mathrm{TP}+\mathrm{TN}}{P+N}$$
(2)$$\mathrm{Sensitivity}=\frac{\mathrm{TP}}{\mathrm{TP}+\mathrm{FN}}\;,$$
(3)$$\mathrm{Specificity}=\frac{\mathrm{TN}}{\mathrm{TN}+\mathrm{FP}}$$

The AUC is the measure of the probability of a randomly chosen positive sample to be higher than a randomly chosen negative sample. The AUC has a range from 0 to 1. When the AUC = 1, the model perfectly distinguishes between positive and negative; conversely, if the prediction of the model is 100% wrong, then its AUC = 0. 

Furthermore, we also used an F1-score, the MCC, and Cohen’s Kappa (*κ*) to determine the model perfectly. 

An F1-score (F1S) conveys the balance between the precision and the recall. It can be formulated as:(4)$$F1S=\frac{\mathrm{TP}}{\mathrm{TP}+\mathrm{FP}+\mathrm{FN}/2}$$

The Matthews correlation coefficient (MCC) is used as a measure of the quality of binary classification. It is a correlation coefficient between the actual and predicted.

The metric of Cohen’s kappa (*κ*) is used to measure the agreement between two raters and is also sensitive to imbalanced datasets.

## 3. Experimental Results

The experiments were executed using Python 3.6 software based on Tensorflow 1.13.1, 16 GB installed RAM, Intel^®^ Core™ i5-8400 CPU @2.80 GHz, and NVIDIA Geforce GTX 1060 6 GB mounted graphic card. Among the 357 patients enrolled (mean age of 48 ± 11.4; BMI index of 23 ± 3.2 kg/m^2^), there were 221 benign and 136 malignant patients. They all had a palpable lump in one or both breasts and had more features of symptoms (nipple retraction or discharge, skin thickening, and lymph node). Women with a diagnosis of benign or malignant also had a tendency for low number of first-degree relatives. A total of 256 out of 731 mammography images had malignant findings as concluded by radiologists.

### 3.1. Performance of the DL Classifiers

The classification performance of the five deep neural networks using mammogram data is shown in Table 2. As is seen from the table, it can be found that the Xception network outperformed considerably the rest of the classifiers with the highest accuracy of 72.5%, sensitivity of 75.7%, and specificity of 70.8%. Additionally, the sensitivity of the VGG16 network was similar to Xception, but the specificity was 48.9%, which was 21.9% less than the best specificity (70.8%). Figure 3 illustrates the ROC curves of all models. The Xception model achieved an F1S of 0.66 and an AUC of 0.79, while the AUC of the rest models was significantly lower. In clinical trials, sensitivity refers to the ability of the test to correctly detect malignant breast cancer, while specificity refers to the ability to eliminate benign lesions. Hence, we selected the Xception network to combine with clinical features in the remaining work.

### 3.2. Performance of the ML Classifiers 

In the clinical dataset, we extracted all the available features related to information about the patient, clinical symptoms, descriptor on ultrasound, and mammogram results (including the size of the lump and BI-RADS categories). The features that had near-zero variance were eliminated: *any family member with breast cancer*, *skin dimpling, and echo pattern (hyperechoic, isoechoic, and mildly hypoechoic)*. We found that there were no continuous variables that had a close correlation (>0.9) with each other; the correlation between weight and BMI is the highest (r = 0.87). Table 3 shows statistically significant differences in remaining variables between benign and malignant outcomes, except for *first-degree family members, the timing of pregnancy, breastfeeding, use of progesterone, and nipple discharge.*

Figure 4 shows the rankings of the number of variables and their performance based on the scores for accuracy and Cohen’s kappa. Through experimental analysis using the RF-RFE method, it was found that there was no significant difference in accuracy when the number of features was increased. Therefore, to simplify the model and optimize performance before applying it to clinical practice, our study selected five clinical variables including *age, palpable lump, nipple retraction, lymph node, and size lump* for imputation in the ML classifiers.

The package *caret* was utilized to implement five different classifiers: k-NN, SVM, ANN, RF, and GBM. Hyperparameter optimization was performed using grid search and five-fold cross-validation. Table 4 compares the performance of selected models using five clinical variables. The best classification performance was obtained with the GBM model in terms of accuracy (81.7%), sensitivity (83.7%), and specificity (78.6%). In addition, this model showed an AUC of 0.84 which was significantly higher than remaining models. Moreover, the results of F1S, MCC, and Kappa also obtained a satisfactory performance. Hence, we selected the GBM classifier to combine with the mammogram-based model.

### 3.3. Performance of the ML-DL Model

We integrated models using mammography images and clinical data to calculate the possibility of either benign or malignant detection for each individual. The input included clinical features and mammography images. The final probability was estimated by: $$\frac{P_{clinical}+P_{image}}{2}$$

The detection breast cancer performance of the ML-DL combining model is observed in Table 5. It shows that adding a clinical-based model in addition to the mammogram-based model provided greater accuracy than only one single model, with the best accuracy reaching 84.5%. The ROC curve in Figure 5 proved that the model built with image and clinical features from the ML-DL model can improve overall performance.

## 4. Discussion

Recent evidence proved that the development of breast image technologies had improved breast cancer survival rates and decreased deaths. Despite these benefits, nearly 30% of breast cancer cases were misdiagnosed. To deal with this problem, our study aimed to investigate the efficiency of integrating clinical information and mammography image in the ML-DL model to improve the estimation of the possibility of cancer. The proposed model was able to achieve acceptable performance in breast cancer detection.

### 4.1. Performance Comparison with the Existing Literature

This study analyzed benign and malignant lesions from mammography images based on five deep neural networks including Xception, VGG16, Resnet-v2, Resnet50, and CNN3. These models were built using a transfer learning technique that fine-tuned the pre-trained deep learning models and added three fully-connected layers for each model. The studies by Benjamin et al. [43] and Samala et al. [44] demonstrated that the transferred CNN model on the mammogram dataset achieved a satisfactory performance for breast tumor detection. In addition, with small-scale mammography image datasets, we used the data augmentation method to increase the number of the relative data, avoid overfitting from training phase, and improve the performance, as some studies have executed [45,46,47]. Our study’s results from deep-learning classifiers show that the Xception network achieved better results than the other models, but the classification performances, in general, were not high. However, it was evaluated as an effective model. To investigate this phenomenon, we compared our proposed method with various best-practice models from the literature summarized in Table 6. Chougrad et al. [48], Mohapatra et al. [49], and Li et al. [50] used Resnet50, VGG16, and Resnet-v2 with over 1500 mammography images. They achieved an accuracy of 97.3%, 65.0%, and 70.0%, respectively. Ting et al. [51] used an available dataset including 221 images which resulted in an accuracy of 74.9%, while Sun et al. [35] collected and analyzed 1874 mammogram images to obtain an accuracy of 82.4%. On the other hand, changing the input size was found as a considerable factor affecting model performance, particularly if the size of breast tissue was decreased by 20% when compared to the initial image size; it could be challenging to discriminate benign or malignant tissue. In our study, all the input images were rescaled into 299 × 299 while Geras et al. [52] completed it with the original image size and the results pointed that the best performance was achieved by using non-rescaled images. Comparing the structure of deep learning models, the Xception, known as a simple model, has a lower number of convolutional layers than other networks at an efficient computational cost. Similar to our Xception model, Chollet et al. also pretrained this network on an ImageNet dataset and obtained an accuracy of 79% [28]. From the above assessments, we concluded that the number of data samples, as well as the quality of the input images, may influence computational performance, and a simple structure model should be the model performed with a small amount of data samples.

To evaluate and compare the ML model performances, we considered the AUC for the analysis as a performance metric for two reasons. First, the distribution in the dataset is imbalanced, therefore criteria such as accuracy can become unreliable for assessing the performance model as it acquires high accuracy for the larger class. Second, the AUC metric is independent of the threshold choice so that keeps a more realistic scenario for the medical application. As the result shows, K-NN had the worst performance (0.76 AUC) in the rest of the models. This could be explained by the fact that K-NN is highly sensitive to data sampling and the number of nearest neighbors. Regarding data imbalance in this study, the major class is benign, so there were about 35.3% of malignant patients misdiagnosed as benign. The same situation occurs for the K-NN model in Boughorbel et al. [53], which also obtained the lowest performance (0.72 AUC). To assess the performance of the model for non-linear data, we compared the SVM and the ANN. It can be observed that the overall results of ANN are slightly higher than SVM because the SVM classifier is not designed to optimize the AUC while ANN is an improvement for an optimization task since it has many parameters and weights that can be optimized for the performance. However, this study has not achieved satisfactory results in the ANN model, which can be explained by the number of input variables as shown in Sepandi et al. [39]. They used input data including mammographic results, demographic, and clinical variables, and achieved the best AUC at 0.95, while the use of only demographic variables in Lee et al. [54] study obtained an AUC of 0.60. Finally, we considered two models from the Boosted Trees algorithm, that is the RF and the GBM. It is shown that the performance of the RF was only higher than the K-NN, although this algorithm is known as a robust classification model. The same situation occurs for the RF. Boughorbel et al. [53] also employed the RF classifier using two different datasets including 32 and 11 clinical variables to obtain an AUC of 0.99 and 0.76, respectively. Their results explained that the RF approach of interpretation involves a logical relation between features, values, and classes. On the other hand, this model is an ensemble of many decision trees, therefore it may be difficult to interpret it for a small dataset. However, the use of boosting trees in the GBM enables a robust model to be outperformed by other models with an 0.84 AUC. Hence, the GBM was identified as an efficient model with small data samples and also as a less complex model, similar to the proposed method by Wang et al. [55].

In addition to calculating the performance of every single model, we analyzed the improvement in combination model performance by incorporating images and clinical variables in the DL-ML model to reduce the possibility of breast cancer misdiagnosis. The model achieved an AUC of 0.88 with 78.1% specificity at 89.7% sensitivity for the detection of breast cancer. Compared with the existing combination model in Moura et al. [19], their results with mammographic features alone obtained 71.5% accuracy, whereas by integrating clinical factors the accuracy was improved to 88.2%. Our study considered the five clinical variables that have proven the importance of these features in evaluating significant improvement between single models and combination models. It was concluded that the detection breast cancer model may be further enhanced by adding clinical information to the image-based model. 

### 4.2. Limitations and Future Developments

Although the combined model exhibited acceptable performance, this study had some limitations. First, a small sample size from our dataset was used to train the proposed model, so the results may not be sufficient to represent the population and limited the model’s performance. Second, the variability in the clinical factor in each population is different. However, we identified the highest contributing features that may lack particular information for a detection objective. Third, many women with benign findings were imported into this study, hence the results may be able to occur due to the biases in the discrimination between benign and malignant.

In the future, the classification performance can be further improved with a larger breast cancer dataset. The results should be validated with different data sources and populations around the world. In addition, it could be noted that using transfer learning models requires large memory and high computational cost, so it may not apply to embedded devices. Due to this issue, we can construct a specific deep learning algorithm for breast image classification with fewer layer architectures; this might highlight potential clinical applications for this algorithm.

## 5. Conclusions

In this paper, we proposed a combined deep-learning and machine-learning model to detect breast cancer that substantially improved performance as compared to a single model. This work demonstrates that combining mammography images and clinical data is advantageous. Four different deep learning classifiers directly learned features from mammography images and were considered along with multiple machine learning classifiers using various clinical variables. Finally, we investigated a combination model by integrating the best two models of those single models and provided an acceptable overall performance. We believe that, in the future, incorporating image data and clinical data can further improve the ML-DL model’s performance. Additionally, the results of this study could be encouraging for the development of a new detection model that can be successful in the application of medical imaging to estimate the probability of breast cancer.

## Figures and Tables

**Figure 1 diagnostics-13-00346-f001:**
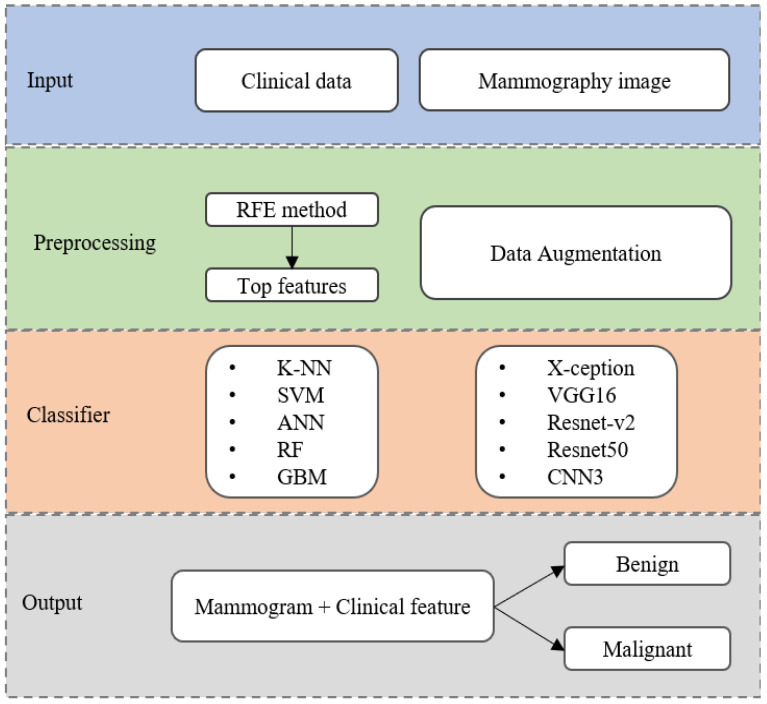
The flowchart of the proposed algorithm using ML-DL model for breast cancer detection.

**Figure 2 diagnostics-13-00346-f002:**
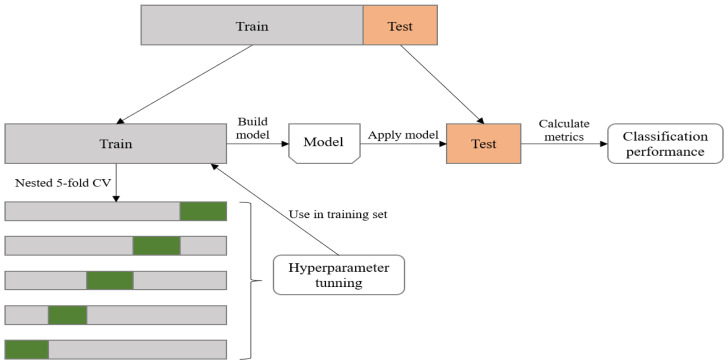
The flowchart of model training, parameter tuning, and performance evaluation.

**Figure 3 diagnostics-13-00346-f003:**
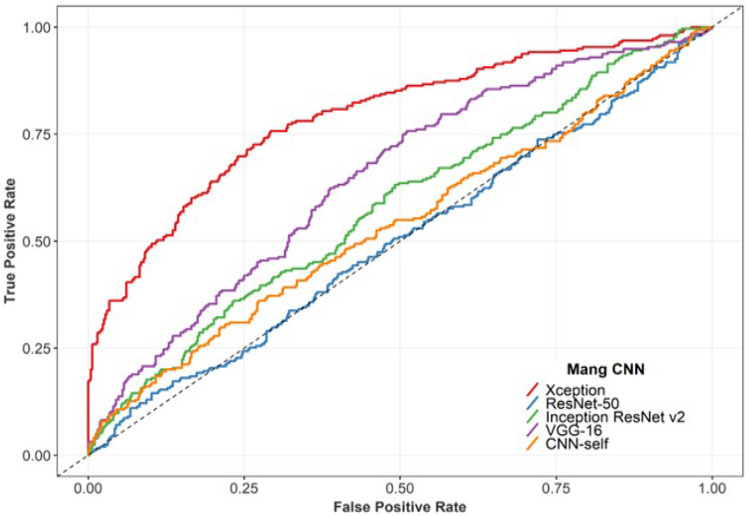
ROC curves of comparison of the classification performances.

**Figure 4 diagnostics-13-00346-f004:**
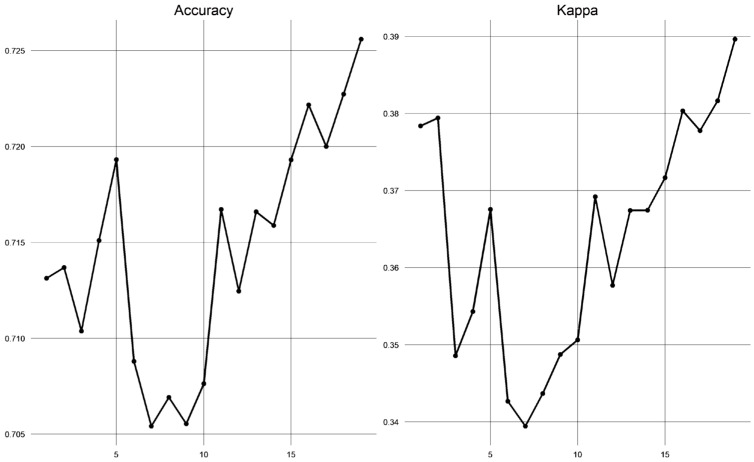
The performance following the number of variables obtained from the RF-RFE method.

**Figure 5 diagnostics-13-00346-f005:**
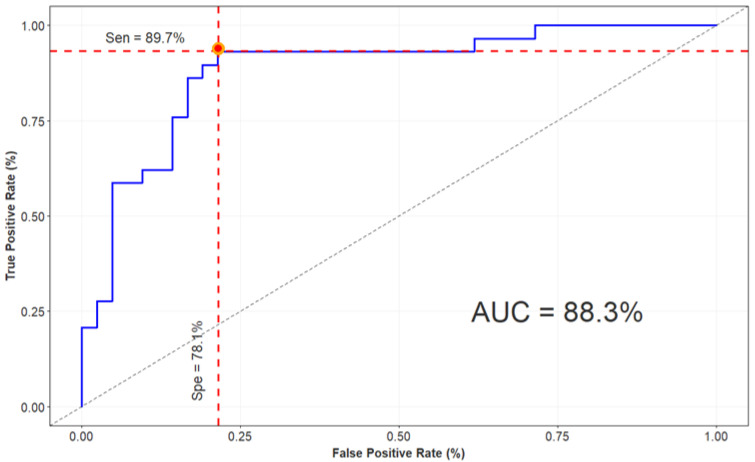
ROC curve of combination model performance.

**Table 1 diagnostics-13-00346-t001:** Tuning hyperparameters in ML classifiers.

Classifier	Caret Package	Fine-Tuned Hyperparameter
k-NN	*knn*	*k* (neighbors)
SVM	*svmRadial*	*σ* (Gaussian kernel)*, C* (Cost)
ANN	*nnet*	*size* (hidden unit), *decay* (weight decay)
RF	*rf*	*mtry* (randomly selected variables)
GBM	*gbm*	*interaction.depth, n.trees, shrinkage*

**Table 2 diagnostics-13-00346-t002:** Classification performance for mammogram data.

Classifiers	Accuracy (%)	Sensitivity (%)	Specificity (%)	AUC	F1S	MCC	Kappa
Xception	72.5 ^#^	75.7	70.8	0.79	0.66	0.45	0.44
VGG16	58.3	75.7	48.9	0.49	0.56	0.24	0.21
Resnet-v2	55.2	63.1	50.9	0.58	0.50	0.13	0.12
Resnet50	50.5	51.0	50.2	0.50	0.42	0.01	0.01
CNN3	53.8	50.6	55.5	0.54	0.43	0.06	0.06

Note: ^#^ denotes the highest accuracy corresponding classifier.

**Table 3 diagnostics-13-00346-t003:** The correlation between clinical variables with outcomes which are benign or malignant.

Variables		Benign	Malignant	*p*-Value
Early menstruation	Yes	4 (1.8%)	10 (7.4%)	0.015
	No	217 (98.2%)	126 (92.6%)	
Late menopause	Yes	43 (19.5%)	58 (42.6%)	<0.001
	No	178 (80.5%)	78 (57.4%)	
Breast skin flaking or thickened	Yes	5 (2.3%)	10 (7.4%)	0.028
	No	216 (97.7%)	126 (92.6%)	
Nipple retraction	Yes	3 (1.4%)	15 (11.0%)	0.001
	No	218 (98.6%)	121 (89.0%)	
Lymph node	Yes	4 (1.8%)	27 (19.9%)	<0.001
	No	217 (98.2%)	109 (80.1%)	
Calcification	Yes	28 (12.7%)	74 (54.4%)	<0.001
	No	193 (87.3%)	62 (45.6%)	
Enhancedvascularity	Yes	31 (14.0%)	77 (56.6%)	<0.001
	No	190 (86.0%)	59 (43.4%)	
Architecturedistortion	Yes	6 (2.7%)	16 (11.8%)	0.001
	No	215 (97.3%)	120 (88.2%)	
Lymph node	Yes	44 (19.9%)	76 (55.9%)	<0.001
	No	177 (80.1%)	60 (44.1%)	
Size lump ^#^		20 (11.1–25.3)	25 (16.6–35.2)	0.002
BI-RADS ^#^		2.86 (2.0–4.0)	4 (4.0–5.0)	<0.001

Note: ^#^ denotes these continuous variables are expressed with mean and standard deviation.

**Table 4 diagnostics-13-00346-t004:** Nested five-fold cross-validation classification performance for clinical data.

Classifiers	Accuracy (%)	Sensitivity (%)	Specificity (%)	AUC	F1S	MCC	Kappa
k-NN	66.2	66.7	64.7	0.76	0.60	0.31	0.30
SVM	73.2	74.5	70.8	0.81	0.67	0.44	0.43
ANN	78.9	81.4	75.0	0.82	0.73	0.55	0.54
RF	69.0	67.9	73.3	0.79	0.64	0.40	0.40
GBM	81.7 ^#^	83.7	78.6	0.84	0.77	0.61	0.60

Note: ^#^ denotes the highest accuracy corresponding classifier.

**Table 5 diagnostics-13-00346-t005:** The classification performance for the combination model.

	Accuracy (%)	Sensitivity (%)	Specificity (%)	AUC
X-ception + GBM	84.5	89.7	78.1	0.88

**Table 6 diagnostics-13-00346-t006:** Performance-based proposed model comparison with existing best-practice algorithms.

Author (Year)	Database (Population)	Category	Classifiers	Classes	Performance
This paper	Privacy dataset (731)	Mammography + clinical data	Xception + GBM	Benign, malignant	Acc: 84.5% AUC: 0.88
Chougrad et al. [48] (2018)	DDSM (5316)	Mammography, mass-lesion classification	VGG16 + Resnet50	Benign, malignant	Acc: 97.3% AUC: 0.98
Mohapatra et al. [49] (2017)	Mini-DDSM (1952)	Mammography	AlexNet + VGG16	Benign, cancer, normal	Acc: 65.0% AUC: 0.72
Li et al. [50] (2020)	Privacy dataset + publicly INbreast (1985)	Mammographic density	Resnet-v2 + CNN	Four BI-RADS categories	Acc: 70% AUC: 0.84
Ting et al. [51] (2019)	MIAS (221)	Mammography	CNN	Benign, malignant, normal	Acc: 74.9% AUC: 0.86
Sun et al. [35] (2017)	FFDM (1874)	Mammogram images with ROIs containing mass extracted	CNN	Benign, malignant	Acc: 82.4% AUC: 0.88
Boughorbel et al. [53] (2016)	METABRIC breast cancer dataset (1981 patients and 11 variables)	Clinical variables and histological	KNN + SVM + Boosted trees	Survived, not survived	AUC: 0.72
Sepandi et al. [39] (2018)	Privacy dataset (655 women and 23 variables)	Demographic and clinical variables	ANN	Benign, malignant	AUC: 0.95
Lee at el. [54] (2015)	Hospital in Korea (4574 cases)	Epidemiological data	SVM + ANN + NB	Case-control	AUC: 0.64
Wang et al. [55] (2014)	Privacy dataset (482 images)	Digital mammography, feature extraction: geometrical, textural	ELM	Image with/without tumor	AUC: 0.85
Moura et al. [19] (2013)	DDSM + BCDR (1762 and 362 instances)	Clinical data + image description	Several ML classifiers	Benign, malignant	AUC: 0.89

Note: DDSM: the Digital Database for Screening Mammography; MIAS: the Mammographic Image Analysis Society Digital Mammogram Database; FFDM: Full-Field Digital Mammography; BCDR: the Breast Cancer Digital Repository; GBM: gradient boosting machine; CNN: convolutional neural network; KNN: *k*-nearest neighbor; ANN: artificial neural network; NB: naïve bayes; ELM: extreme learning machine; ML: machine learning.

## Data Availability

The Oncology Hospital Ho Chi Minh City database is not publicly available due to privacy and ethical issues.

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
