# Peer review of "Development of an Artificial Intelligence-Based Breast Cancer Detection Model by Combining Mammograms and Medical Health Records"

_diagnostics, 2023, doi:10.3390/diagnostics13030346_

Round 1
Reviewer 1 Report
This paper discusses the performance of various traditional- and advanced methods for breast cancer detection in medical images. To achieve it, they combined mammography images and medical health records. The paper should be improved before publication, and my major comments are as follows:
1) Novelty and Contribution are unclear and must be emphasized in the paper. As a suggestion, the authors may open a new subsection and describe the Novelty and Contribution of the paper.
2) In the second section, the authors should provide more information about the dataset, such as the images' size, channel size, etc.
3) The related work section should be improved by including the most recent papers related to breast cancer detection and classification. For instance,
MITNET: a novel dataset and a two-stage deep learning approach for mitosis recognition in whole slide images of breast cancer tissue
Breast Cancer Detection in Mammography Images Using Deep Convolutional Neural Networks and Fuzzy Ensemble Modeling Techniques
Automatic Detection and Classification of Mammograms Using Improved Extreme Learning Machine with Deep Learning
Classification of Breast Cancer Histopathological Images Using DenseNet and Transfer Learning
Fuzzy ensemble of deep learning models using choquet fuzzy integral, coalition game and information theory for breast cancer histology classification
Automated diagnosis of breast cancer using multi-modal datasets: A deep convolution neural network based approach
An automated deep learning based mitotic cell detection and recognition in whole slide invasive breast cancer tissue images
Multi- class classification of breast cancer abnormalities using Deep Convolutional Neural Network (CNN)
4) The authors should also include an equation for AUC.
5) The authors included parameters description for machine learning methods, but I couldn't find a description of parameters (e.g. optimization, learning rate, dropout rate, etc.) for deep learning methods. The authors should include this information in the paper.
Reviewer 2 Report
The abstract needs quantification. The authors combined to different methods to detect a breast cancer mode which is a good research. However, the conversion of medical records in to numerical values are not reported completely. Tree diagram may be added in the introduction. F1 Score, MCC and Kappa are also analyzed. Certain clarification is required for discussion section. The conclusion may be changed. The references are enough.
Round 2
Reviewer 1 Report
The authors addressed all my comments and improved the manuscript extensively. The paper is now ready for publication.